# Are Nurses Aware of Their Contribution to the Antibiotic Stewardship Programme? A Mixed-Method Study from Qatar

**DOI:** 10.3390/healthcare12151516

**Published:** 2024-07-31

**Authors:** Nesiya Hassan, Albara Mohammad Ali Alomari, Jibin Kunjavara, Kalpana Singh, George V. Joy, Kamaruddeen Mannethodi, Badriya Al Lenjawi

**Affiliations:** Nursing and Midwifery Research Department, Hamad Medical Corporation, Doha 00974, Qatar

**Keywords:** antibiotic stewardship, antibiotic stewardship programme, multidrug-resistant microorganisms, antimicrobial stewardship

## Abstract

The antibiotic stewardship programme (ASP) is a new concept initiated by WHO, but nurses are not yet ready to adopt the program. The training and empowerment of nurses are the best strategies for enhancing their knowledge and engagement in ASP. This mixed-method study was used to assess perceived roles and barriers of nurses’ involvement in ASP. An online survey was conducted among 420 clinical nurses to identify their role, and 23 individual interviews were performed among nurses and infection control practitioners to explore the barriers and recommendations to overcome the identified barriers. The majority of the nurses agreed with the sixteen identified roles in ASP, of which ‘antibiotic dosing and de-escalation’ (82.61%), ‘IV to PO conversion of antibiotic, outpatient antibiotic therapy’ (85.23%), and ‘outpatient management, long-term care, readmission’ of the patients (81.19%) had the lowest agreement from the participants. The major themes generated through the qualitative interviews were a lack of knowledge about ASP, poor communication between multidisciplinary teams, lack of opportunity and multidisciplinary engagement, lack of formal education and training about ASP, lack of ASP competency and defined roles in policy, role conflict or power/position, availability of resources, and lack of protected time. Nurses play an integral role in the successful implementation of antibiotic stewardship programs. The empowerment of nurses will help them to adopt the unique role in ASP. Nurses can significantly contribute to antibiotic stewardship efforts and improve patient outcomes through addressing these challenges.

## 1. Introduction

Antimicrobial resistance is a global health concern and threat to human life. The possible strategy to overcome this issue is to judiciously conserve the effectiveness of existing antibiotic resources to prevent the development of multidrug-resistant microorganisms (MDRO), which is known as antibiotic stewardship (AS). The World Health Organization (WHO) defines AS as “The careful and responsible management of something entrusted to one’s care” [1]. The fundamental aim of antimicrobial stewardship involves organized measures to promote the optimized use of appropriate antimicrobials to reduce antimicrobial resistance and the spread of diseases due to multidrug-resistant organisms [2]. The proper use of antibiotics is primarily focused on patient safety and improving clinical outcomes. Disruption of the microbiome can lead to major health issues, such as sepsis, that can threaten the lives of patients.

An integrative approach of the American Nurses Association (ANA) and Centers for Disease Control and Prevention (CDC) was to initiate nurse-driven antibiotic stewardship programmes. They acknowledged the critical role of nurses in antibiotic stewardship programmes; however, a wide gap in knowledge, especially regarding the concept of antibiotic stewardship, knowledge of microbiology, and use of various antibiotics [3], existed among the nurses.

Despite the important role that nurses hold in ASPs, they may not recognize it as a vital part of ASPs, and they may perceive that they have a lack of knowledge, skills, or authority to satisfy this role [4]. The lack of knowledge about antibiotic stewardship among nurses has been highlighted by many studies [5,6]. Systematic reviews support that nurses have a lack of awareness related to AS, but many of the components related to ASPs are practiced in their daily clinical practices, which include education and monitoring the safety of antibiotics [7]. The nurse’s perception of medical hierarchical tradition is the major obstacle that keeps them away from active enrolment in ASPs [8,9]. The other barriers identified in the literature include the additional workload, a lack of support from the medical team, knowledge gaps, a lack of confidence and experience, competency issues, a lack of resources, time constraints, and finally a lack of assurance in their opinions [6,10,11]. Nurses play a leading role in infection control and prevention programmes such as central line-associated bloodstream infections (CLABSI), catheter-associated urinary tract infections (CAUTI), and ventilator-associated pneumonia (VAP) through effective implementation of all elements of the bundle approach. The outcome of these programs has shown that nurses’ empowerment is crucial to the success of the programme [12].

According to the Joint Commission International (JCI) announcement in 2017, one of the top priorities is to establish antimicrobial stewardship standards in all organizations [13]. Moreover, the WHO considers ASP as one of the three pillars of an integrated approach to strengthening health systems. A study conducted in Qatar during 2019 revealed that the core elements of ASPs are moving with international guidelines. However, the key barrier identified in this study was the lack of quality, frequency, and consistency of education regarding ASPs [14]. Currently, the organization has an Anti-Microbial Stewardship (AMS) committee at the corporate and facility level. However, the antimicrobial stewardship clinical policy of the organization [15] demands only minimal roles and responsibilities from clinical nurses. Many of the elements of the ASP are routinely performed by nurses without any recognition. Moreover, there are limited studies addressing this issue from a Middle Eastern perspective. In this context, the current study conducted in Qatar aims to assess the perceived roles and barriers of clinical nurses in active engagement in ASPs using a mixed-method approach.

## 2. Materials and Methods

### 2.1. Study Design

The present study used a mixed-method research design to collect the data from the participants. In this convergent parallel design, the role of clinical nurses in ASP was measured quantitatively, whereas the qualitative part explores the barriers of nurses in stewardship programmes.

### 2.2. Sample Size

The sample size was calculated in the quantitative study based on the prevalence of perceived knowledge (65%) regarding antibiotic stewardship [9], with a 95% confidence interval and 5% precision, and considering the 10% of incomplete questionnaires, the total sample size was 400. The qualitative part of the study was to conduct 23 semi-structured interviews to reach saturation.

### 2.3. Sample and Population

The study was conducted in the largest health organization in Qatar. The population is composed of licensed clinical nurses working in all facilities and administering antibiotics as part of their routine duty. The study excluded nurses who exclusively performed administrative duties in the clinical setting. A purposive sampling method was used to recruit the participants in this study, and the qualitative part included 19 clinical nurses as well as four infection control practitioners to gather the required data.

### 2.4. Data Collection

The quantitative data were collected through an online survey among the clinical nurses working from 15 June 2022 to 3 July 2022 including fourteen facilities under the corporation. The eligible participants were invited to the study through an open invitation using a hospital email address.

The qualitative interview, nineteen clinical nurses and four infection control practitioners were interviewed from those who expressed their intention to participate. An individual semi-structured interview was conducted between 30 October 2022 and 16 July 2023. A suitable interview time was scheduled according to the convenience of the participants. Consent was secured from the participants before the interview and the average duration was 25–30 min. The clinical nurses were recruited to achieve a diversity of experience, and in order to explore the difference in working exposure, the deliberate strategy was used to select the nurses from emergency departments, different critical care units, and inpatient units including medical, surgical, and paediatric units. After conducting three interviews with the clinical nurses, the interview transcripts were reviewed by the research supervisor (AB) who provided advice and feedback on interviewing techniques to obtain deeper responses from participants on perceived barriers of ASP. The interviewer (NH) had ample experience in the antibiotic stewardship programme and in research.

### 2.5. Data Sources/Measures

The study used an adapted tool from Olans RD after obtaining permission. The tool has been widely used in the literature for assessing the role of nurses in ASP [8,9,16]. The questionnaire has 16 questions in four subdomains including, patient admission (5 items), daily clinical progress monitoring (3 items), patient safety and quality monitoring (4 items), and clinical progress/patient education/discharge (4 items). The participants were requested to choose their best responses for each item. The response was marked as “Not at all responsible” and “responsible” according to their perception. A descriptive statistic was used to summarize and determine the sample characteristics and their agreement and disagreement over the roles of nurses in ASP.

The qualitative part focused on exploring the nurses’ perceived barriers and recommendations for active engagement in ASP through individual semi-structured interviews using an interview guide. The semi-structured interview was set with a pre-determined key topic of ASP based on the literature review [17,18]. This was explored through open-ended questions to encourage open discussion and the generation of themes regarding perceived barriers and recommendations in active engagement in ASP. The data were collected and transcribed verbatim in Microsoft Teams, and the audio was recorded with permission of the participants.

The transcription was openly coded and verified by the investigator (NH and AB) and analysed through the process of horizontalization [19]. Two members of the team checked and completed summarizing themes discovered during the interview process. A thematic approach was used to analyse the qualitative data. Extraction of data involved the coding frame development and categorizing the data through a combination of similar concepts raised by the participants. During the data analysis, 30 significant statements were revealed by the participants. These statements were examined for contextual meaning and clustered into different themes, which can support the evidence of what the perceived barriers are, and recommendations for clinical nurses’ active engagement in the ASP program. To ensure the accuracy of the data analysis, a peer debriefing was conducted after completing fifteen interviews with team members (AB and NH). Although 17 interviews with clinical nurses and three interviews with infection control practitioners were believed to have reached saturation, two additional interviews with clinical nurses and one with the infection control practitioner were conducted to confirm the data saturation.

## 3. Results

### 3.1. Characteristics of the Study Participants

A total of 427 nurses accessed the online survey in which seven surveys were excluded from the analysis due to incomplete responses. In the quantitative part, the female-to-male ratio of the participants was 5:2, and most of the participants (57.61%) were between 31 and 40 years of age. The majority of the respondents hold bachelor’s degrees (89.05%) and 41.42% of the participants have one to five years of experience in the organization. Most of the study participants were working in critical care and inpatient units accounting for 36.42 and 38.33%, respectively. Moreover, many of the participants (76.9%) did not receive any formal training in ASP (Table 1).

Similarly, in the qualitative part, most of the participants were females (74%) and the majority of the interviewees belonged to the 31–40 years age group. The majority of the participants (95.7%) hold bachelor’s degree in nursing, and the vast majority of the participants (78.2%) have more than 5 years of experience in the organization. Most of the respondents (78.2%) claimed that they did not receive any training regarding ASP.

The participants’ responses regarding the perceived roles of nurses in ASPs have been summarized in Table 2. The nurses’ roles are categorized into four subdomains, namely patient admission, daily clinical progress monitoring, patient safety and quality monitoring, and clinical progress/patient education/discharge. The roles of “accurate assessment of allergy history” and “early and appropriate cultures” accounted for the participants’ greatest levels of agreement in the patient admission domain, with 98.09% and 97.85%, respectively. Similarly, there was 99.04% and 97.85% agreement on the tasks of progress monitoring and reporting and ‘reviewing preliminary culture results and discussing with the treating physician’. Ninety percent of participants consented to each role in patient safety and quality monitoring domains, which include adverse events reporting, change in patient condition, final culture report and antibiotic adjustment, and antibiotic resistance identification. Nurses’ role in patient education had the greatest agreement (95.95%) from participants compared to other roles in clinical progress/patient education/discharge.

The qualitative part focused on the barriers and recommendations of nurses’ engagement in ASP, and several subthemes emerged from the interview. Thematic analysis of the interview data identified nine sub-themes reflecting areas of barriers in nurses’ participation in ASP programmes (Table 3).

#### 3.1.1. Theme 1—Lack of Knowledge about ASP

The participants had a certain extent of familiarity with common types of antibiotics used for their patients. They emphasized the need for a comprehensive understanding of antibiotics, their actions, adverse effects, proper monitoring methods, special precautions, etc. This theme was agreed with the majority of the participants (N1, N2, N3, N4, N5, N6, N7, N8, N10, N11, N13, N14, N15, N18, IP1, IP2, and IP3).

“*We need a proper idea about every antibiotic, and how long it usually needs to be administered. We are not concentrating on what we are administering, its action, and the side effects when we must assess this antibiotic.*”(N1)

Moreover, nurses felt that if they were aware of the clinical indication of the antibiotic, it might enhance their confidence in ASP.

The majority of the participants understood ASP’s aim to optimize antibiotic use, reduce overuse, and ensure appropriate prescriptions. There was a consensus among participants that antibiotic stewardship will help to improve patient outcomes, minimize microbial resistance, and prevent unnecessary antibiotic usage. The potential consequences of inadequate stewardship, which include prolonged hospital stays and additional health care costs, highlight the importance of effective implementation and active engagement in stewardship initiatives.

“*Selecting of most appropriate antibiotic for the specific diseases based on blood culture results not a trailed with some other unnecessary antibiotics and so the patient will receive the right antibiotic.*”(N2)

The participants underscored the significance of appropriate antibiotic selection based on factors such as blood culture and sensitivity reports and patients’ co-existing clinical conditions.

“*The antibiotic should be given to the patient according to the sensitivity and it has to be continued over the course of time, the antibiotic has to be changed according to the sensitivity, and antibiotic they should not stop or should not added with another antibiotic.*”(N3)

“*Antibiotics should be used very cautiously because the antibiotics usage can lead to the resistant organisms to develop in the bloodstreams of the patient.*”(N5)

The participants shared their challenges regarding their level of knowledge and confidence related to various antibiotics used in clinical practice.

“*I am confident in teaching my patients regarding common antibiotics, but I’m not sure regarding all the antibiotics, so I’m not confident enough to educate the patient.*”(N7)

Some of the nurses felt that they have a lack of familiarity with ASP; N8 stated that “*Many of the nurses do not know about antibiotic stewardship*”. As a result, participants avoided active engagement in ASPs.

“*Lack of knowledge regarding this stewardship program may be pulling me backward from active participation in this program. We don’t know much regarding how to escalate the antibiotic, the generation of antibiotics, etc. if I want to discuss something, I need to know more about that topic.*”(N15)

“*I feel that nurses are not much confident because their base is not concrete on the awareness of antibiotic stewardship program itself.*”(IP1)

There is a recognized need for advanced knowledge to engage in discussions and escalate antibiotic-related topics, indicating a desire to enhance understanding and confidence among nurses.

#### 3.1.2. Theme 2—Poor Communication between Multidisciplinary Teams

Nurses face many challenges in communications related to ASPs and antibiotics. Hence, they have a sense of uncertainty regarding the programme due to the lack of dissemination of information about the ASP. Participants N2, N3, N5, N7, N8, N9, N10, N11, N12, N14, N15, N17, and IP1 agreed on this theme.

“*Most of our staff is not that much aware of this program and we are not included in any educational program or meetings.*”(N8)

“*What to do and what not to do in the antibiotic stewardship program should be clear. There is no team, there is no coordination in this team. In the antibiotic stewardship program, there is no regular meeting or regular education, or no pamphlet or educational materials were distributed.*”(N3)

Participants reported that coordination and communication are absent within the ASP, indicating that “*we don’t have any committee collaborating with the pharmacist, physician, and nurses”* (N7). This highlighted the need for establishing collaborative teams, expanding the ASP committee with clinical nurses, as stated by N9. “*If I get training, I can communicate and educate my other colleagues and will encourage others also to participate in this program*”.

As IP 1 recommended, “*The committee itself has to extend their communication effectively to the nurse, so I think something is lacking*”.

#### 3.1.3. Theme 3—Lack of Opportunity and Multidisciplinary Team Engagement

The collaboration and coordination between the nurses and healthcare workers in the ASP are limited. The theme was agreed upon by participants N6, N7, N8, N9, N10, N12 N15, N17, and IP1.

As N12 stated, “*There is no special committee, or any nursing representative involved in the ASP committee*”. Nurses expressed a desire to be included in discussions, training, and updates related to the ASP; “*The nurses must know regarding ASP because we are the main part of administering and monitoring the side effects of the antibiotics.*” (N10)

However, in current practice, participants stated “*Doctors are only consulting with the clinical pharmacist and ID team every time, no nurses involved in this discussion*” (N6). Nurses emphasized their first-hand knowledge and importance as the primary source of information about patients, advocating for their inclusion in discussions and decision-making processes.

To “*make a team to work for antibiotics stewardship program*” (N8) that includes physicians, nurses, and other team members is suggested to ensure collaborative involvement and representation in the ASP.

#### 3.1.4. Theme 4—Lack of Formal Education and Training about ASP

There was a consensus among nurses about the absence of structured training programmes dedicated to the ASP. This represented a major obstacle for the clinical nurses to embed the basic principles of ASP in daily clinical practice. It was agreed by most of the participants, N1, N2, N3, N4, N5, N6, N7, N8, N9, N10, N11, N12, N14, N15, N18, N19, and IP4.

“*I didn’t receive any formal training.*”(N4)

“*I didn’t receive any training, any courses, conference, workshop or online training, in the stewardship program.*”(N8)

“*We don’t have an opportunity to attend proper training or any course regarding ASP.*”(N15)

The lack of workshops or teaching activities related to stewardship further emphasizes the need for structured training opportunities.

#### 3.1.5. Theme 5—Lack of ASP Competency and Defined Roles in Policy

The lack of ASP competency and specified roles in existing policy hinder the nurses’ successful adoption into an ASP. Participants N3, N4, N6, N7, N8, N11, N13, N14, N16, and IP1 agreed on this theme. Nurses also expressed that

“*There is no mandatory competency for the nurses in ASP.*”(N7)

“*There are no specific competencies and self-learning programs about stewardship.*”(N14)

Nurses express their belief that having specific competencies in ASP would improve their commitment to updating ASP-specific policies and guidelines. Competency-based assessments and training programs may enhance nurses’ knowledge and skills in ASP.

#### 3.1.6. Theme 6—Role Conflict or Power/Position

The participants expressed that they were confused about their participation in ASPs as “*Doctors never discuss with the nurses and we will get rare chances to discuss with the doctors*” (N14), and N10 identified that “*Sometimes nurses are not aware why these antibiotics are stopped*”. There is a sense that nurses’ opinions and concerns are not valued or recognized by physicians, leading to a limited role for nurses in the ASP. Moreover, nurses perceived a lack of interest, acceptance, and value given to their concerns and suggestions by some doctors. This attitude contributes to a sense of frustration and a perception of limited influence in the ASP. The theme was approved by participants N1, N2, N4, N6, N7, N9, N10, N13, N14, N15, IP1, and IP3.

“*We will think they did not give any value to our concerns. Even a small percentage are accepting or concentrating about our concerns.*”(N14)

Nurses feel that their roles are limited to administering antibiotics and monitoring side effects.

“*This program is owned by physician and pharmacy. I think we are lacking the opportunity to be in this program.*”(N2)

Nurses demanded collaborative teamwork, committees, and recognition of their expertise and perspectives in decision-making processes related to antibiotic stewardship.

“*We need to involve the nurses in the main role of stewardship program because they are the primary source of information in any program like sepsis and CUTI.*”(N13)

#### 3.1.7. Theme 7—Resources

Nurses emphasized the vital role of clinical pharmacists in educating both clinical nurses and patients about antibiotics, ensuring their safety and confidence.

This was agreed upon by participants N1, N2, N3, N4, N5, N6 N7, N8, N9, N10, N11, N12, N13, N14, N15, IP1, and IP2.

“*The clinical pharmacists are more competent to educate nursing staff, patients, and their families about the antibiotics. The presence of clinical pharmacists makes us feels more safer.*”(N1)

Moreover, nurses highlighted the use of databases, drug libraries, and online references to obtain accurate and up-to-date information for antibiotic administration.

“*It is important to access the information and resources for antibiotic administration.*”(N8)

#### 3.1.8. Theme 8—Lack of Protected Time

Nurses expressed the difficulties in allocating time and involvement in ASP due to the nature of their work, especially in areas demanding higher levels of care. Participants N1, N2, N3, N5, N11, N12, N14, N15, N19, and IP1 agreed on this theme.

“*Practice of ASP in emergency situations is a little bit chaotic.*”(N1)

They underlined the need for monitoring the use of antibiotics, reviewing individual cases, and discussing the guidelines at the unit level in order to help to improve their confidence.

“*Lack of time, because we are in critical care area with busy schedules.*”(N15)

The ASP activities might be unintentionally neglected due to a heavy clinical workload.

#### 3.1.9. Theme 9—Recommendations and Future Opportunities

The participants highlighted the “*importance of training, education, and competency development*” (N15) in preparing the nurses for ASPs. Nurses suggest that periodic assessment of knowledge enhances their engagement in updating antibiotic stewardship protocol. Active involvement and communication of nurses in the antibiotic stewardship programme is one of the major recommendations from the participants.

“*Include nurses or representatives in ASP so we will be aware of new changes.*”(N12)

They emphasized the need for link nurses in ASP committees, which should be initiated by leaders, and protected time for attending stewardship activities. To ensure high-quality and consistent care, the participants emphasized the need for a systematic approach to assess the achievement of goals and ensure the effectiveness of stewardship practices.

“*If there is any monitoring tool for ASP, it will be easy to know whether we achieved the goal or not.*” (N11)

Nurses stressed the importance of having E-Drug formularies attached to drug orders, providing detailed information, precautions, and proper administration of antibiotics. Creating sustained awareness among nurses is major challenge in ASPs, which can be resolved through periodic awareness sections as IP 2 suggested:

“*The facilities can celebrate one day as antibiotic stewardship awareness day to create more awareness.*”

## 4. Discussion

In this study, we explored the role and barriers of clinical nurses in antibiotic stewardship programmes using a mixed-method design. The study results showed a high level of agreement among nurses towards different roles in ASP. This revealed that nurses are confident in their roles and contribute to the success of antibiotic stewardship programmes. Nurses are performing many activities in ASPs, but their efforts and contributions are not recognized and integrated into clinical practice, which is confirmed in the present study [16]. Céline Bridey et al. (France, 2023) reported that nurses have a positive attitude towards taking over a more prominent role in ASP, whereas they demanded additional training, modifications in the existing legislative framework, and support from multidisciplinary teamwork [20]. The results of this study further support these findings and provide insights into nurses’ perceptions of their roles in ASPs within the Middle Eastern context.

World Health Organization’s guidelines [21] on antimicrobial stewardships highlight the importance of promoting the appropriate use of antimicrobials to improve patient outcomes and quality of care across the continuum of care. In alignment with this, the current study also reveals that the accurate assessment of allergy history and early and appropriate cultures were identified as essential components of ASPs, as well as the key role of nurses. The roles of progress monitoring and reporting, as well as reviewing preliminary culture results and discussing them with the treating physician, received high agreement from our participants. These findings are in line with previous studies that have emphasized the importance of nurses’ active involvement in monitoring patient progress and collaborating with healthcare providers in making informed treatment decisions which include questioning the necessity of urinary culture, encouraging the prescribers to switch from IV to oral antibiotics; initiating review after 48 h after antibiotics are commenced and de-escalating from broad-spectrum to narrow-spectrum antibiotics [22,23].

Patient education is a critical component of ASPs, as it promotes adherence to treatment regimens, empowers patients to participate in their care, helps prevent the misuse or overuse of antibiotics and decreases the risk of AMR. Nurses should be well equipped to provide tailored education to patients and their families, addressing misconceptions and ensuring a proper understanding of antibiotic therapy [8]. Additionally, nurses’ participation in clinical progress monitoring and discharge planning can contribute to seamless transitions of care, continuity of treatment, and assessing the adverse events associated with antimicrobial use [8]. The current study highlighted the nurses’ agreement on their involvement in patient education and discharge planning.

The study also exposed existing barriers among nurses to their active participation in ASP. A lack of updated knowledge and confidence in ASP was the major barrier recognized by the participants which was consistent with previous research [24,25,26]. Formal training can improve nurses’ knowledge and confidence in ASPs and enable them to work effectively with the multidisciplinary team [23]. In our study, infection control practitioners suggested that formal education and training can help in the successful adoption of ASPs among clinical nurses. Furthermore, antimicrobial stewardship is the key element in reducing AMR.

A multidisciplinary approach is considered the best practice for the success of AMS; however, in ASPs, the nurses are not included in all activities [27] including committees and clinical rounds [8]. Furthermore, the nurses’ opinions were not actively sought [8] and they experienced pushback from physicians [9,28]. This is consistent with our study’s findings; the participants expressed their concerns that they are not part of the current ASP team, clinical meetings, and ASP committee reviews. Suggestions included regular updates and communication, as well as continued education and integration of nurse’s involvement in the successful implementation of ASP strategies [22,25]. Lack of time and failure to prioritize AMS activities due to clinical workload were commonly mentioned as barriers to the integration into clinical practice [4,29]. These findings are consistent with our study, where the participants perceived that their clinical role and patient care are more imperative than participating in ASPs. The heavy clinical workload impedes nurses from taking any additional roles in participating in ASPs. Moreover, the nurses perceived that many roles recommended in ASPs were lacking relevance or beyond their scope of traditionally held nursing responsibilities [4,26]. The perceived hierarchies and professional role conflicts can hinder the communication between the health care teams [28], which is consistent with our study findings, the participants experienced confusion and conflict in their role in ASPs.

In the current study, nurses mentioned the availability of resources, including online, and support from physicians and clinical pharmacists in their facilities. Nurses are confident to reach out with their concerns and questions regarding ASPs. In contradiction to a previous study, that the participants were updating their knowledge through clinical experience [22]. Nurses are in a unique position to educate patients and families about antimicrobial stewardship and the healthcare organization should focus on building confidence in nurses.

### Strength and Limitations

To the best of knowledge, this is the first study to address the nurse’s role and barriers in ASPs in Qatar. The sample was composed of staff who are working in various units, such as critical care areas, emergency departments, and inpatient units, and thus, the results portrayed reflect the views and opinions of the nursing staff. The study highlights the opinions of nurses in Qatar. Furthermore, the study was based on representation from all fourteen facilities working under the corporation. However, there are limitations in this study. Although we included different levels of nurses such as clinical nurses, charge nurses, head nurses, and infection control practitioners, the study did not involve any multi-disciplinary healthcare professionals. Also, there might be chances of selection bias during face-to-face interviews, and unique practices within the facilities might affect their opinions.

## 5. Conclusions

The nurses are ready to integrate antimicrobial stewardship initiatives into the nursing practice, even if their contribution is often not recognized. The nurses play a critical role in preventing antimicrobial resistance including taking a major practice role in the antimicrobial stewardship programme, ensuring prescribing antibiotics matches patients’ allergic history, collecting and following up with culture results, and educating the patient and family regarding the importance of hand hygiene in line with local policy. Nurses also play a significant role in promoting the principles of antimicrobial stewardship in their roles advising patients. Every healthcare organization must expand and accommodate the ASP committee with nurses, as well as ensuring that nurses are getting opportunities to advance their knowledge in antimicrobial stewardship and remain up to date in clinical practice.

## Figures and Tables

**Table 1 healthcare-12-01516-t001:** Socio-demographic and professional characteristic of the participants.

	Survey Participants	Interview Participants
Characteristics	Frequency (%) (N = 420)	Frequency (%) (N = 23)
Sex		
Female	300 (71.42)	17 (74.00)
Male	120 (28.57)	6 (26.00)
Age in years		
21–30	43 (10.24)	1 (4.30)
31–40	242 (57.61)	15(65.30)
41–50	112 (26.67)	4 (17.40)
>50	23 (5.48)	3 (13.00)
Highest Qualification		
Diploma	46 (10.95)	1 (4.30)
BSN *	333 (79.29)	15 (65.30)
MSN **	41 (9.76)	7 (30.40)
Years of experience in this organization		
<1	25 (5.96)	0 (0.00)
1–5	174 (41.42)	5 (21.80)
6–10	96 (22.86)	8 (34.80)
11–15	66 (15.71)	3 (13.00)
>15	59 (14.05)	7 (30.40)
Clinical area		
Emergency department	60 (14.29)	5 (21.80)
Inpatient department	161 (38.33)	4 (17.40)
Critical care	153 (36.42)	8 (34.80)
Outpatient department	17 (4.05)	2 (8.60)
Others	29 (7.00)	4 (17.40)
Training in Antibiotic stewardship programme		
Yes	97 (23.10)	5 (21.80)
No	323 (76.90)	18 (78.20)

* Bachelor of Nursing; ** Master of Nursing.

**Table 2 healthcare-12-01516-t002:** Participant’s agreement on identified role in the Antibiotic stewardship programme.

Domain	Item No	Identified Roles	Rank	Response of the Participants (N = 420) (%)	
				Agree	Disagree
Patient admission	1	Triage and appropriate isolation measures	4	402 (95.71)	18 (4.29)
	2	Accurate assessment of allergy history	1	412 (98.09)	8 (1.90)
	3	Early and appropriate cultures	2	411 (97.85)	9 (2.14)
	4	Timely initiation of antibiotics	3	404 (96.19)	16 (3.81)
	5	Medication reconciliation	5	377 (89.76)	43 (10.24)
Daily (24 h) clinical progress monitoring	6	Progress monitor and report	1	416 (99.04)	4 (0.95)
	7	Reviewing preliminary culture results and discussing with the treating physician	2	411 (97.85)	9 (2.14)
	8	Antibiotic dosing and de-escalation	3	347 (82.61)	73 (17.38)
Patient safety and quality monitoring	9	Adverse events reporting	1	416 (99.04)	4 (0.95)
	10	Change in patient condition	2	414 (98.57)	6 (1.43)
	11	Final culture report and antibiotic adjustment	4	379 (90.23)	41 (9.76)
	12	Antibiotic resistance identification	3	397 (94.52)	23 (5.48)
Clinical progress/patient education/discharge	13	IV * to PO ** conversion of antibiotic, outpatient antibiotic therapy	3	358 (85.23)	62 (14.76)
	14	Patient education	1	403 (95.95)	7 (1.67)
	15	Length of stay	2	386 (91.90)	34 (8.10)
	16	Outpatient management, long-term care, readmission	4	341 (81.19)	79 (18.81)

* Intravenous; ** Per-oral.

**Table 3 healthcare-12-01516-t003:** Barriers and facilitators to nurses in active role in the antimicrobial stewardship programme.

Themes	Subthemes
Clinical nurses experienced barriers in participating ASP	Lack of knowledge about ASP
	Poor communication between multiple teams
	Lack of opportunity and multidisciplinary team engagement
	Lack of formal education and training about ASP
	Lack of ASP competency and defined roles in policy
	Role conflict or power/position
	Resources
	Lack of protected time
Recommendation for the clinical nurses in future engagement ASP	Education and training
	ASP competency
	Defined roles in ASP policy
	Nurse champions in ASP
	Enrolment in ASP committee
	Protected time for participation in ASP
	Periodic meetings and update
	Celebrate ASP awareness day

## Data Availability

Data are contained within the article.

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
