# Peer review of "Are Nurses Aware of Their Contribution to the Antibiotic Stewardship Programme? A Mixed-Method Study from Qatar"

_healthcare, 2024, doi:10.3390/healthcare12151516_

Round 1

Reviewer 1 Report

Comments and Suggestions for Authors

“Are nurses aware of their contribution to the Antibiotic Stewardship Program? A mixed-method study from Qatar” by Hassan et al. 

In this paper the authors describe the contribution of nurses to antimicrobial stewardship program in Qatar. In the Global Action Plan adopted by the World Health Organization, there should be multiple contributors for successful implementation of Antimicrobial Stewardship Program. This includes physicians, pharmacists as well as nurses and the patients. However, as described in this paper, nurses have largely been left out of this program in Qatar. The conclusions are based on surveys and interviews of a large number of clinical nurses. Since nurses have a greater interaction with patients admitted in hospitals than physicians and pharmacists, they are in a unique position to make a significant contribution in the implementation of the program. So this paper addresses an important drawback of the program. The paper has been mostly written properly, however, there are numerous grammatical errors in the whole manuscript. Representative comments by nurses have been presented as quotes.  The very poor English language in these quotes puts the nurses in a negative light and has the opposite effect of what the authors are trying to emphasize. 

            I also have the following comments on the manuscript, mostly regarding incorrect English. 

Line 47: Change “role that nurses hold in ASPs, nurses may not recognize” to “role that nurses hold in ASPs, they may not recognize”

Line 104: Change “The consent” to “Consent”

Line 106-109: “The clinical nurses were recruited to achieve diversity of experience, and deliberate strategy to explore the difference in working exposure the nurses were selected from emergency departments, different critical care units, and inpatient units including medical, surgical, and paediatric units.” The sentence is grammatically problematic. Please modify. Suggestion: “The clinical nurses were recruited to achieve diversity of experience, and in order to explore the difference in working exposure, the deliberate strategy was to select the nurses from emergency departments, different critical care units, and inpatient units including medical, surgical, and paediatric units”

Line 110, 133, 143 and 447: “AB” Not clear what AB stands for.  It does not match the name of any author in the list. Most likely it is for Albara Mohammad Ali Alomari, but why is it abbreviated as AB?

Line 116: Change “was widely used” to “has been widely used”

Line 119: Change “(3 items)” to “(3 items),”

Line 135: Change “were checked” to “checked”

Line 136: Change “used to analysis” to either “used to analyse” or “used for analysis of”

Line 149-151: “This section may be divided by subheadings. It should provide a concise and precise description of the experimental results, their interpretation, as well as the experimental conclusions that can be drawn.” This appears to be more like an editorial directive than like a presentation of results.

Line 187: Add space before “Lack”

Line 192: “how long usually needs to be administered” to “how long it usually needs to be administered

Line 195: Change “Nurses” to “nurses”

Line 197: Change “ASP programs” to “ASP’s” The abbreviation, ASP includes the word “program”.

Line 211: “and antibiotic they should not stop or should not added with another antibiotic” Although these statements are written as quotes, they should still be grammatically correct. All participants in the survey were certainly not speaking wrong English.

Line 213: “can lead to the resistant organisms to develop in the bloodstreams” Same comment as for line 211.

Line 222: “pulling backward me” Same comment as for line 211.

Line 226: “I feel is like that” Same comment as for line 211.

Line 226: “because there base is not concrete” Same comment as for line 211.

Line 227: “awareness of on antibiotic” Same comment as for line 211.

Line 231: Add space before “Poor”

Line 232: Change “communicating related to” to “communications related to”

Line 238: “there is no coordination like this team” Same comment as for line 211.

Line 239: “there is no pamphlet or educational materials were distributed” Same comment as for line 211.

Line 248: Change “up to the nurse” to “to the nurse”

Line 256: “consulting with this clinical pharmacist” I guess, the authors mean “consulting with the clinical pharmacists” Also, same comment as for line 211

Line 269: Change “don’t” to “didn’t” Also, same comment as for line 211.

Line 283: Add space before “Role”

Line 295: “providing patient care makes than valuable contributors” This is grammatically incorrect and it is not even clear what the authors are trying to say.

Line 301: “because the information or any programs which is gathering from the nurses.” Not clear what this means.Line 303: Add space before “Resources”

Line 303: “Resources challenges” Not clear what this means.  Is “challenges” here a noun or verb? If it is a noun, add a comma after “resources”

Line 305: “creating a sense of safety and confidence.” From the language, it is not clear what is “creating a sense”

Line 306-308: “The clinical pharmacists are more competent to educate both of us as a staff and then and the patients and their families about the antibiotics presence of clinical pharmacists feels safer for us and the patient also as well” Not clear what this means. It does not appear to be written in English. Also, same comment as for line 211.

Line 315: Change “Practice ASP” to “Practice of ASP” Also, same comment as for line 211.

Line 317: Change “will help” to “in order to help”

Line 318: Change “Llack” to “Lack”

Line 318: Change “Llack of time because we are in critical care area, busy schedules.” to “Lack of time, because we are in critical care area with busy schedules.” Also, same comment as for line 211.

Line 333: “A monitoring tool, it will be easy to know” Not clear what this means. Please correct.

Line 338: “at least like a once in a quarter” Grammatically incorrect. Please correct.

Line 344: Change “in their roles are contribute” to “in their roles and contribute”

Line 345: “The findings from this study conflict with previous studies that……clinical studies” The studies mentioned here are observations of which study? This study or the cited study? What is the conflict?

Line 347: Change “et al.;” to “et al.,”

Line 378: Change “has identified the” to “has identified that”

Line 380: Change “has identified formal training can” to “has identified that formal training can”

Line 382: Add space after “team”

Line 408: Add space after “.”

Line 414: “study that which” Not clear what this means.

Comments on the Quality of English Language

Quality of English is poor in some places. Representative comments by nurses have been presented as quotes.  The very poor English language in these quotes puts the nurses in a negative light and has the opposite effect of what the authors are trying to emphasize. Although these statements are written as quotes, they should still be grammatically correct. All participants in the survey were certainly not speaking wrong English.

Reviewer 2 Report

Comments and Suggestions for Authors

Dear Authors,

ASPs are extremely important - thanks for this research.

I have highlighted the text and added comments on the left side. If there are two highlights for a line of text, the first comment relates to the first highlight. 

I recommend that you early on mention which country the research was conducted in, as the wide gap in knowledge you describe is very much country-specific. In some countries, there are very advanced nurse-led ASPs.

Pg 3; Line 109-112: I am concerned that using probing and prompting techniques can lead to biasing or influencing the responses from nurses. This is very unusual in this kind of interview.

 Pg 3; Line 127: The semi-structured interview was set with a pre-determined key topic of ASP based on the literature review. However, you have not previously mentioned the literature review as part of the methods.

Comments on the Quality of English Language

This manuscript needs a lot of updating. Corrections and suggestions have been made

Author Response

I appreciated your valuable comments. Please see the attachment.

Reviewer 3 Report

Comments and Suggestions for Authors

This paper attempts to discern attitudes, values, barriers and awareness of the ASB.  To that end this mixed-methods research succeeds.

The Introduction reviews the relevant background literature and some of the issues associated with nurse’s concerns.  The specific aims/purpose of the study are not well-articulated.  The last paragraph of the Introduction should address this issue.

The Materials and Methods section carefully explains the survey methods and the qualitative interviews. I would have liked to know the “pre-determined” key topics identified by for the semi-structured interviews. The method for the qualitative analysis adequately describes the process undertaken by the authors.

The descriptive analysis, Table 1, presents information allowing the reader to better understand the sample’s characteristics.  Likewise, Table 2 identifies aspects of the role of nurses in the ASP. However, the survey results would be more interesting if several questions related to barriers and facilitators were asked.  This would allow some comparison and validation with the interviews.  There appears to be adequate literature to accomplish this.

The nine themes and the methods employed to identify cover a range of important issues.  Some respondents are denoted in this section.  I would like to know how many interview participants mentioned each theme. This would give some quantification of the extent of agreement with the themes.

The Discussion is far-ranging and needs to be more focused and less word intensive. I had some difficulty in ascertaining exactly what the study revealed and why the results are important. Part of my difficulty is the connection between the survey and the interviews.  Also, paragraphs, Lines 353 – 417, should be reworked so that the resulting paragraphs focus on one thought. It is not clear exactly how the survey results, Table 1, are incorporated in the Discussion.

Author Response

(The authors gave the same response as above.)

Round 2

Reviewer 2 Report

Comments and Suggestions for Authors

Dear Authors,

Thanks! The manuscript reads well. I have no further comments.

Author Response

Thank you. I appreciated your valuable comments.